# Blood Flow Restriction Training and Its Use in Rehabilitation After Anterior Cruciate Ligament Reconstruction: A Systematic Review and Meta-Analysis

**DOI:** 10.3390/jcm13206265

**Published:** 2024-10-20

**Authors:** Jamaal Butt, Zubair Ahmed

**Affiliations:** 1Department of Inflammation and Ageing, School of Infection, Inflammation and Immunology, College of Medicine and Health, University of Birmingham, Edgbaston, Birmingham B15 2TT, UK; jyb4@student.le.ac.uk; 2Department of Neurosurgery, University Hospitals Birmingham NHS Trust, Mindelsohn Way, Edgbaston, Birmingham B15 2GW, UK

**Keywords:** anterior cruciate ligament, blood flow restriction training, rehabilitation, ACLR, functional recovery

## Abstract

**Background/Objectives**: Anterior cruciate ligament (ACL) reconstruction (ACLR) is often followed by significant muscle atrophy and subsequent loss of strength. Blood flow restriction training (BFRT) has recently emerged as a potential mode of rehabilitation to mitigate these effects. The goal of this systematic review was to evaluate the efficacy of BFRT in functional recovery when compared to traditional rehabilitation methods. **Methods**: A literature review was conducted across July and August 2024 using multiple databases that reported randomised controlled trials comparing BFRT to traditional rehabilitation methods. Primary outcomes were changes to thigh muscle mass and knee extensor/flexor strength with secondary outcomes consisting of patient-reported functional measures (IKDC and Lysholm scores). The RoB-2 tool was used to assess the risk of bias. **Results**: Eight studies met the inclusion criteria; however, substantial heterogeneity prevented a meta-analysis being conducted for the primary outcomes. Three out of the five studies measuring muscle mass reported significant (*p* < 0.05) findings favouring BFRT. There was variation amongst the strength improvements, but BFRT was generally favoured over the control. Meta analysis of the secondary outcomes showed significant improvements (*p* < 0.05) favouring BFRT despite moderate heterogeneity. **Conclusions**: BFRT shows promise for maintaining muscle mass and improving patient reported outcomes following ACL reconstruction. However, the high risk of bias limits the strength of these conclusions. Further high-quality research needs to be conducted to establish optimal BFRT protocols for this cohort and to determine if BFRT has a place in ACL rehabilitation.

## 1. Introduction

Anterior cruciate ligament (ACL) injuries affect 20,000 people annually [1] and present a significant level of morbidity for patients. Surgical reconstruction (ACL reconstruction (ACLR)) with subsequent rehabilitation is often the preferred method of treatment to improve long-term outcomes. However, following completion of treatment, individuals are often left with atrophic changes and strength deficits of up to 40% when compared to the contralateral leg [2,3]. This atrophy has been identified as unresponsive to traditional strength training and rehabilitation, leading to significantly reduced strength and functioning of their quadriceps muscle and extensor aspect of the knee, which in turn impacts the success of rehabilitation programs along with affecting daily functioning [4,5]. Residual strength deficits following rehabilitation have been linked to increased risks of reinjury and post traumatic arthritis.

A broad range of mechanisms have been proposed to be behind the reason for these atrophic changes. Some of these reasons include: (1) disuse related anabolic resistance due to upregulation of receptors that facilitate muscle breakdown and a decrease in muscle synthesis [4,6]; (2) reduction in circulating satellite cells which are thought to have a role in facilitating muscle remodelling following injury [4,7]; (3) neuromuscular changes resulting from disruption of the innervation of the muscle due to a decrease in the number of neuromuscular junctions that are present [4,5]. Current modes of rehabilitation focus on early weight bearing: the utilisation of braces at fixed levels of extension or following rehabilitation protocols such as straight leg raises, walking up flights of stairs and heel slides. Optimising rehabilitation programs to account for the residual deficits in quadriceps strength has been challenging but blood flow restriction training (BFRT) has begun to emerge as an intervention that can help to address all these issues [8,9].

BFRT is a method of strength training that was first pioneered in the 1960s by Y. Sato in Japan and later refined in the 1990s after extensive testing, with protocols still being developed and improved to this day [9]. The training involves partial vaso-occlusion of a limb utilising an inflatable pressure cuff and training with weights of around 30% of a patient’s one rep max. This method of training helps to simulate and elicit strength and hypertrophy gains distally: results that would typically only be seen with traditional strength training using around 70% of an individual’s one rep max [9,10,11]. Where blood flow restriction training comes into its own in a clinical environment is its application in groups of patients who are unable to tolerate traditional strength training for any reason. In line with the original premise behind the development of BFRT, it is incredibly suitable for those who have just been injured such an in an ACL injury (part of the original development was centred around a meniscal knee injury suffered by Y. Sato) [10] or in the immediate postoperative period following an ACLR, where heavy loading of the joint runs the risk of disrupting any grafts that have been put in place [12].

The current state of literature with regards to BFRT focuses primarily on its use to improve muscle strength and size as well as examining its effects on patient-reported outcomes in contrast against traditional methods of rehabilitation [9,13]. The application of BFRT in ACLR rehabilitation has been gathering increasing attention over recent years, with some results demonstrating improved quadriceps strength compared to traditional rehabilitation protocols [14]. The most recent systematic review on this topic was published in 2023 with the literature search of this study containing all studies up to 2021 [15]. It highlighted issues with heterogenous training protocols in the field and a lack of consistency in measuring outcomes, concluding that the benefit of BFRT over standard rehabilitation was not clear. As the body of literature continues to grow, and with BFRT becoming more popular across social media and in clinical practice, there is a need for a comprehensive synthesis of the current evidence to help guide clinical decision-making and future research directions. There is also a general lack of standardised protocols and inconsistent outcome measures used in different studies, and hence we sought to synthesis the evidence to understand which outcome measures could be used to best demonstrate the efficacy of BFRT.

The aim of this study was to understand the impact of BFRT on maintaining and improving muscle mass and leg strength following an ACLR through a systematic review and meta-analysis of the published literature. The intention of this paper was to compile findings from recently published randomised controlled trials along with those that had already been included in previous systematic reviews on the same topic to determine if there had been any change in the literature on whether BFRT had a place in everyday clinical practice. The population examined in this review were adults who had an ACLR due to ACL injury. We compared those who used BFRT as part of their rehabilitation protocol following ACLR against those who performed standard rehabilitation without the use of BFRT following ACLR. Primary outcomes of interest were muscle size and strength of the upper leg following rehabilitation. Secondary outcomes consisted of patient-reported functional measures, which were also analysed where possible to determine if patients noticed an improvement in their day-to-day functioning. The functional measures of choice were the Lysholm and IKDC knee scores, as the former is specific to ligamentous injury within the knee and the latter is used as a general measure of knee function [16,17]. Both the Lysholm and KDC scores are utilised extensively across the literature with regard to ACLR, and using them in this case would allow for the results of this study to be placed within the context of the wider field. More importantly, they have both been validated for use in a post ACLR cohort and are able to detect changes in function over time [18,19].

## 2. Materials and Methods

### 2.1. Search Strategy

The search strategy was developed and conducted in accordance with Preferred Reporting Items for Systematic reviews and Meta-Analyses (PRISMA) [20] recommendations and guidelines. However, the study was not registered in a public repository. The databases used for the search were Pubmed, Ovid Medline (R), Embase and Web of Science. Search terms consisted of ‘ACL’, ‘Injury’, and ‘Blood flow restriction’. Whilst broad search terms were used to help ensure that the maximum amount of literature was returned, the search was conducted multiple times across all databases to refine them sufficiently. The final search took place in August 2024, utilising Pubmed, Ovid Medline (R), Embase and Web of Science. The same search terms were used across all three databases and consisted of ‘Anterior cruciate ligament OR ACL’ and ‘Blood flow restriction OR BFR’ and ‘injury’. The word ‘injury’ was included to help exclude papers that examined degenerative ACL tears, as this was not within the scope of this review.

### 2.2. Inclusion and Exclusion Criteria

Articles were eligible to be included in the study if they were in English, or an English translation was available, measured BFRT as a method of rehabilitation following ACLR, and compared the intervention group against another that followed either the standard rehabilitation protocol for that geographical area or a protocol that was the same as the intervention group but without the BFRT component. The study design had to be that of a randomised control trial (RCT). Studies were excluded if they contained participants who were under the age of 16 or examined sport-specific athletic populations as these groups of people are not representative of the general population and would have led to the exclusion of recreational athletes as well as creating a heterogenous sample. Papers were also excluded if they did not measure at least one of the outcomes of interest or if the study design was not of a randomised controlled trial. Studies were not filtered by date to allow for data from all relevant studies until August 2024 to be included. Additional exclusion criteria consisted of conference abstracts, systematic reviews, cohort studies, or other forms of prospective or retrospective studies where randomisation of the groups did not take place.

### 2.3. Data Collection

Following the searches being run, each database was manually screened to identify appropriate articles by title and abstract. Covidence was then used to aid the data collection process [21]. Results returned from the searches on the respective databases were uploaded to make tracking and compilation of search results more efficient. Following initial screening by the primary author (J.B.), the selected articles were then sent to the co-author (Z.A.) for verification and to allow for any issues that arose to be resolved via discussion. Following this, articles then underwent a full text review to ensure that they fulfilled all the exclusion criteria.

### 2.4. Data Extraction

Extraction of data from eligible studies was conducted by the primary author (J.B) with verification of appropriateness once again being conducted by the co-author (Z.A). Extracted data were made up of the following: The type of blood flow restriction that had been used, which included the type of cuff and brand name where appropriate as well as occlusion pressure, which was either expressed as a percentage or pressure/mmHg. Number of participants and their distribution across different arms. Length of rehabilitation and where possible, how long after surgery rehabilitation began as well as when data were collected, which was provided as either weeks from surgery or weeks from the beginning of rehabilitation.

Primary outcome-related data that were collected for both control and intervention groups consisted of changes in femur/thigh muscle mass, the mechanism of measurement and where possible, data were included from before and after the rehabilitation program. The secondary primary outcome for which data were collected was the change in leg strength with respect to flexion and extension of the knee. Data were collected on the mechanism of measurement and where possible, data from before and after the invention were also recorded. Where primary outcome data were provided in graph form without clear tabulation of the numerical values, Claude 3.5 AI software (Anthropic, San Francisco, CA, USA) was used to extract the mean and standard deviation from the graphs. Clause 3.5 is a well-known AI model that has been programmed and developed for use in statistical analysis and data extraction. Appropriate prompts were written, and the reported results were then visually validated against the original graph [22]. Authors were contacted for missing data, but no responses were received during the time frame of this review. Data related to secondary outcomes of interest for both control and intervention groups consisted of changes to either the IKDC or Lysholm functional scores, and where possible the scores from before and after the rehabilitation program. For all outcomes, *p* value data were collected to enable the significance of results to be determined

### 2.5. Risk of Bias Assessment

All the studies included in this review were RCTs, so it was appropriate to use the Risk of Bias 2 (RoB-2) tool, as recommended by Cochrane, to assess the quality of the studies [23]. Articles were screened by both authors (J.B and Z.A) independently and then the results were shared, and any discrepancies between the two authors were discussed. The RoB-2 tool consists of five domains to assess the risk of bias across the study with each domain being rated as having a low, unclear, or high risk of bias with an overall risk of bias for each study then being generated. The use of this tool enabled conclusions to be drawn about the quality of the research that took place.

### 2.6. Statistical Analysis

For each of the outcomes that had data extracted, the mean and standard deviation for results from both the control and intervention groups were collected, and then where there was sufficient homogeneity between at least three studies, a meta-analysis using RevMan 4.0 software (Cochrane Collaboration, London, UK) was conducted. A random effects model was used, and the mean difference and 95% confidence interval were reported with heterogeneity of the results assessed using the I^2^, Chi^2^, and Tau^2^ statistics. A random effects model was chosen over fixed effects due to the heterogeneity in the studies, where the random effects model can cope with both within-study and between-study variances. Forest plots were used to visualise the results compared to the null line. Where data were not able to be statistically analysed due to heterogeneity between studies, a narrative synthesis of the data was conducted instead.

## 3. Results

### 3.1. Study Selection

After conducting the search in line with PRISMA recommendations, 204 articles were returned from searches conducted across the three databases. Search results were then uploaded to Covidence (www.covidence.org; Melbourne, Victoria, Australia), where nine articles were automatically identified as duplicates and removed, and 93 articles were marked as ineligible for review by the tool, as they were not randomised controlled trials. The eligibility tool was at the time of writing, in early access, so a manual validation of the excluded search results was also conducted to ensure that no records had been inappropriately excluded. This process resulted in 102 articles remaining to be screened by title and abstract. Following this second stage of screening, 10 records were sought for full text retrieval; one study was not able to be accessed [24], which left nine studies for full text screening. Out of the nine studies, two were excluded, one for not assessing the outcomes of interest [25] and another for not following the desired study design [26]. Alongside this, one article was identified from citation searching and was deemed to be appropriate following a full text screen [27]. This resulted in a total of eight studies that were assessed as being appropriate for inclusion in this review as can be seen in Figure 1.

### 3.2. Study Characteristics

In line with the inclusion/exclusion criteria, all of the included studies were RCTs (Table 1). Full data collection tables are provided as Appendix A. Seven out of the eight studies [27,28,29,30,31,32,33] assessed the efficacy of BFRT in ACLR rehabilitation against a group who followed the same rehabilitation program but without the BFRT component whereas one study [34] had both groups follow a standard NHS rehabilitation program but added an extra strength training component on top, comparing low-load BFRT resistance training against high-load resistance training without BFRT. The study designed the two strength training programs in line with recommendations on the different types of strength training. There was an array of occlusion pressures across the different studies. Three studies [28,29,34] utilised an 80% arterial occlusion pressure (AOP) and one study [33] used a 40% AOP. Another study had two intervention groups, which used 40% and 80% AOP, respectively [30]. The remaining three studies [27,31,32] used fixed levels of cuff inflation to provide blood flow restriction, varying from 130 mmHg up to 180 mmHg. The length of rehabilitation programs varied from 14 days [27] up to 16 weeks [32] with three of the studies having programs that were twelve weeks long [28,29,33] and two studies having programs that were eight weeks long [30,34]. In each study, the participants ages were closely matched; the youngest group of participants had a mean age of 24.9 (±7.4) years old [27], and the oldest group of participants had a mean age of 41.1 (±9.8) years of age [29]. One study did not provide a detailed breakdown of patient ages across groups [30].

### 3.3. Risk of Bias

As stated in the methods, the RoB-2 tool was used to assess bias within the studies, and as a way to gauge the quality of the research that had been conducted. The outcome of the tool demonstrates that there was a moderate to high risk of bias across all studies with five out of the eight studies [27,28,31,33,34] assessed as having a high overall risk of bias and three of the eight studies [29,30,32] assessed as there being some concerns over the risk of bias (Figure 2). There were concerns across all five domains. Three studies [27,28,33] had unclear randomisation protocols, with one of these failing to address randomisation entirely [33]. Concerns about adherence to the intended intervention were raised in four of the studies [27,31,32,33], as they utilised hand pumped tourniquets outside of a clinical environment, which means that patients may not have sufficiently inflated the blood pressure cuffs if they found them uncomfortable to deal with. Additionally, one of the four studies allowed for patients to increase the pressure as they saw fit which means that the AOP would not have been consistent across the intervention group [27]. Three studies [27,31,33] were assessed as having a high risk of bias in the domain that assessed the inclusion of all the appropriate final outcome data; there was the omission of the generation of toque and patient reported outcomes, both of which had been assessed prior to the intervention. Due to the inability to truly blind the patients to the intervention, there were some concerns over the risk of bias in the measurement of the patient reported functional outcomes (IKDC, Lysholm), as the participants filled these out themselves, and they may have been unduly influenced by the group that they had been assigned to [27,29,30,32,33]. The risk of bias in the selection of reported results follows on from the missing outcome data in the sense that it was not made clear whether those studies simply did not record the data post completion of the rehabilitation program or whether the data was recorded but not reported [27,31].

### 3.4. Primary Ouctome Data

Five out of the eight studies [27,28,30,31,33] included in this review reported on differences in thigh/femur muscle mass over the course of the rehabilitation program (Table 2). Significant heterogeneity amongst the methods of measurement meant that it was not possible to conduct a meta-analysis comparing the data. Measurement methods included a DEXA scan [28], ultrasound of the vastus lateralis [34], and cross-sectional area of the quadriceps [27,31]. Three of the five studies [28,30,31] found there to be a statistically significant change in muscle mass between the control and BFR groups, indicating that BFRT was effective at increasing muscle mass over the standard rehabilitation techniques. Two of the studies [28,31] reported *p* < 0.01, and the remaining study [30] reported *p* < 0.05. The remaining two studies [27,34] did not report any statistically significant differences, and no trend was seen between the intervention and control group in this regard. Subgroup or sensitivity analyses were not performed due to the high risk of bias which impacts on all data interpretation in this systematic review.

### 3.5. Secondary Outcome Data

A meta-analysis was performed for both functional scores. A random effects model was used for both analyses with the intervention being favoured in both cases. The IKDC demonstrated a total mean difference of 5.90 favouring the intervention (*p* = 0.01), whilst moderate heterogeneity is displayed with I^2^ = 49% and Chi^2^ = 5.90 (Figure 3). Two out of the four studies [28,30] report positive effects favouring the intervention with the remaining two studies [33,34] demonstrating a non-significant outcome. There was an overall positive effect favouring the intervention.

The aggregate Lysholm score demonstrated a total mean difference of 6.75 favouring the intervention (*p* = 0.02), whilst moderate to substantial heterogeneity was once again displayed with I^2^ = 56% and Chi^2^ = 4.55 (Figure 4). All three of the studies demonstrated trends, favouring the interventions, but two out of the three studies [33,34] reported a non-significant outcome. There was an overall positive effect favouring the intervention.

## 4. Discussion

The primary purpose of this review was to determine whether there is a benefit to using BFRT as part of the rehabilitation process for patients, following ACLR with the goal to restore muscle mass and strength at a quicker rate than traditional rehabilitation methods. Our analysis of eight studies revealed several key findings, but it also highlighted substantial limitations and helped to identify areas for future research.

Overall, it is difficult to draw a solid conclusion with regards to increases in muscle mass. The theoretical basis for BFRT was supported [35] by the results, which demonstrated significant findings in three out of the five studies. There is a large amount of heterogeneity in protocols and outcome measurements with each study reporting muscle size differently. With regards to strength training, the picture is a bit clearer due to more uniform methods of measurement. BFRT generally resulted in a stronger knee extension at higher angular velocities but not at lower velocities, with its effectiveness appearing to be influenced by longer rehabilitation time periods and higher levels of occlusion [8]. Whilst generalised training protocols exist for use on healthy individuals, this highlights a need for the development of a more standardised protocol that can be used for ACLR patients.

The secondary outcomes provided a picture that is consistent with the wider literature, in that BFRT improves functional outcomes. This is a particularly pertinent point as patients’ self-perceptions are a key factor that contribute to the success of rehabilitation [36,37]. However, the same issues arise in terms of the application of these results, in that moderate to substantial amounts of heterogeneity, as confirmed by the meta-analysis make it difficult to draw conclusions. In addition to this, the inability to blind patients may have influenced these outcomes.

To place this review in the context of the wider literature, the previous review into the efficacy of BFRT following ACLR must be considered. As touched upon in the introduction, issues with heterogenous study protocols and variability in outcome measurement made it difficult to draw clear conclusions [15]. This review demonstrated that these issues remain and tarnish any conclusions that can be drawn. It was shown that although it was possible to observe a trend that generally favoured the use of BFRT in order to improve upper leg muscle and size, results were not always significant. Where significant trends were seen, as was the case with the secondary outcomes, large amounts of heterogeneity highlight the need for more standardised investigatory protocols within this field.

A recurring theme throughout this review has been the issue of heterogeneity amongst the included studies. Heterogeneity is present at all levels and highlights the lack of standardised protocols in this area, also alluding to why there is inconsistency in the reporting of statistically significant results. Table 1 and Section 3.2 both show how each study included in this review adopted a different approach. For example, a total of four studies used the same AOP of 80% [28,29,30,34], and of these four, the same brand of cuff and occlusion pressure was only used in two instances [28,34]. Another source of heterogeneity was found within the specific rehabilitation protocols. Table 1 further demonstrates how studies did not begin rehabilitation at consistent times in the post operative period and also that the length of rehabilitation varied between studies. With regards to specific rehabilitation protocols, only four studies adopted the same range of repetitions per exercise [28,30,33,34]. However, within these four studies, there did not appear to be consistency with the specific exercises that were performed or the intensity. Whilst it is promising that the more recently published papers have shown the beginning of a more homogeneous nature of protocols, there remains a need to take this even further to enable stronger conclusions to be drawn from this body of literature.

Although not directly in the scope of this review, at the beginning, reference was made to the hidden factors that affect atrophy following ACLR. Two of the studies recorded data on this; one on muscular recruitment [33] and another on inflammatory markers [30]. Whilst not within the objectives that were laid out, it points to another direction that future research can take—to help aid understanding of the more precise mechanisms of the role of BFRT in reducing muscle atrophy [37].

### Limitations

A significant limitation of this review is the moderate to high risk of bias, as assessed by the RoB-2 tool, that is prevalent across all the selected studies. Issues such as unclear randomisation, the inability to truly blind in most cases, and potential selective reporting of results, raise concerns about the reliability of these findings. The small sample size of approximately 27 in each study further limit the ability to apply these findings to the general population. Whilst the risk of bias analysis did not prevent our outcomes reporting or performing meta-analysis with pooled data from the outcome, the interpretation of the findings was severely affected. A high risk of bias means that any significant effects described in the meta-analysis must be interpreted with caution. As discussed at multiple points throughout this review another limitation was the heterogeneity across protocols, outcome measures and time frames, which prevented a meta-analysis from being conducted but is also representative of the evolving state of the literature.

Despite these limitations, the trends that favour BFRT, particularly in maintaining muscle mass and in functional outcomes, suggest that it could have a place in ACLR rehabilitation protocols. It could be used as an adjunct in early therapy when loading of the joint with heavy weight is contraindicated; however, its implementation should be approached with care [38,39]. Individual patient characteristics should be considered, and questions remain surrounding the optimal timing and intensity of BFRT, evidenced by the studies included in this review. Future research should focus on recruiting larger cohorts with clear reporting of processes throughout the study, such as clear discussion of randomization processes and the reporting of results. Standardisation of BFRT protocols in this context would facilitate more robust meta-analysis taking place, allowing for more definitive conclusions to be drawn. Longer follow-up times would also allow for more information to be collected on functional outcomes once patients return to daily life.

A potential limitation of our systematic review is that we did not register the study into an appropriate public repository. This was an oversight on our part, and it is something that we also recommend to avoid potential bias. Having the review registered allows for any deviations from the original protocol to be documented. In addition, the exclusion of studies focusing on athletes of a single discipline to focus on recovery in the general population may mean that our results are not generalizable to this group of people who are at most risk of ACLR.

## 5. Conclusions

Whilst BFRT shows promise as an additional tool for ACLR rehabilitation, a large amount of research still needs to be conducted before any conclusions can be drawn about its suitability for implementation in clinical practice. Unfortunately, this echoes the conclusions of previous systematic reviews in this area, and whilst specific training protocols do exist, these are targeted towards healthy individuals. Further research needs to be conducted into surgery-specific cohorts as well as into a broader range of patients of different ages and backgrounds, as the cohorts for all of the included studies included young and generally only health individuals. This is in part due to the nature of ACL injuries but leads to the exclusion of patients who may already have medical conditions and represent situations where the use of BFRT is not as clear cut. As stated at the beginning of this review, the field is one that is still rapidly evolving, and with new literature constantly being published we can hope that a clearer picture is painted in the not-too-distant future.

## Figures and Tables

**Figure 1 jcm-13-06265-f001:**
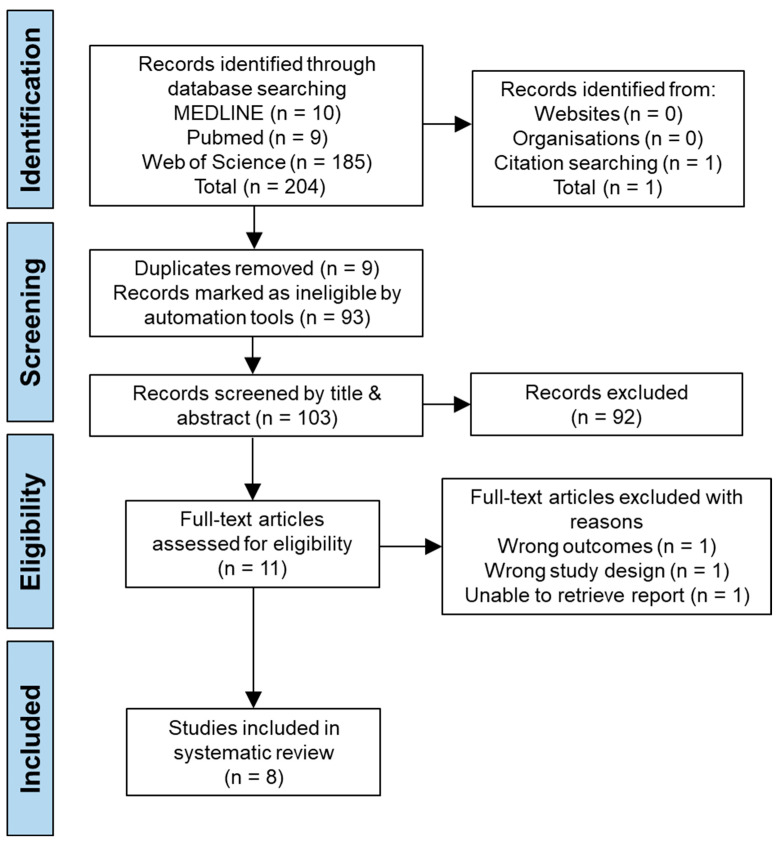
PRISMA flow diagram.

**Figure 2 jcm-13-06265-f002:**
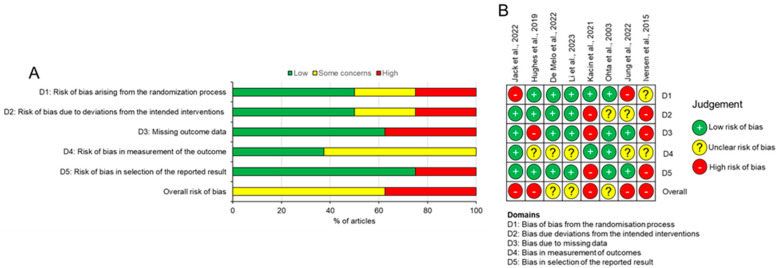
Risk of bias assessment. (**A**) summary of risk of bias in all studies. (**B**) Risk of bias in individual studies for each domain [9,27,28,29,30,31,32,33].

**Figure 3 jcm-13-06265-f003:**
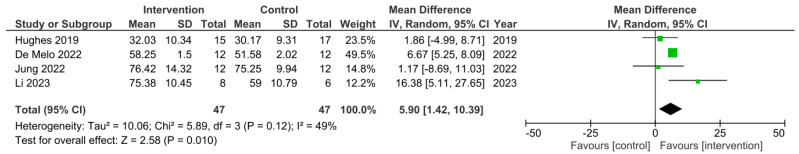
Meta-analysis of secondary outcome data for IKDC scores in appropriate studies [9,29,30,33].

**Figure 4 jcm-13-06265-f004:**
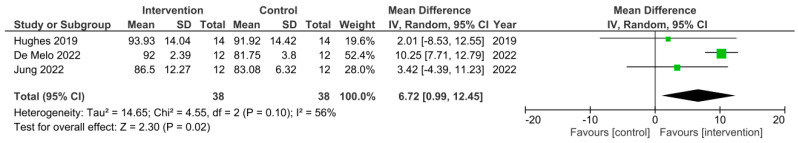
Meta-analysis of secondary outcome data for Lysholm scores in appropriate studies [9,29,33].

**Table 1 jcm-13-06265-t001:** Characteristics of the included studies.

Authors	Country	Occlusion Type	N Number	Gender	Participant Age/Years	Length of Rehabilitation/Weeks	Timing of Data Collection After Initial Evaluation/ Weeks
Jack et al., 2022 [28]	USA	Delfi Medical automated torniquet at 80% arterial limb occlusion pressure around proximal thigh	Control–17 BFR–15	Control–M7/F8 BFR–M12/F5	Control–24.1 (±7.2) BFR–28.1 (±7.2)	12 weeks (biweekly sessions) (beginning within 7 days post op)	8 and 12 weeks from beginning training
Hughes et al., 2019 [9]	UK	Delfi Medical automated torniquet at 80% arterial limb occlusion pressure around proximal thigh	Heavy Load Resistance Training–14 BFR–14	Heavy Load Resistance Training–M10/F2 BFR–M7/F5	Heavy Load Resistance Training–29 (±7) BFR–29 (±7)	8 weeks (biweekly sessions) (beginning at 2 weeks post op)	9 weeks from the beginning of training
De Melo et al., 2022 [29]	Brazil	Cuff Scientific Leg^®^–WCS 80% arterial limb occlusion pressure around proximal thigh	Control–12 BFR–12	Control–M9/F3 BFR–M8/F4	Control–39.6 (±10.8) BFR–41.1 (±9.8)	12 weeks (biweekly sessions)	4/8/12 weeks post op
Li et al., 2023 [30]	China	Air band Bluetooth pressurised device at 40% occlusion pressure and 80% occlusion pressure	Control–6 40% BFR–9 80% BFR–8	Gender split across groups was not provided	18–40 years old but the specific split was not detailed	8 weeks (biweekly session) (beginning at least 8 weeks post op)	8 weeks from the beginning of training
Kacin et al., 2021 [31]	Slovenia	BFR GROUP Double-chamber pneumatic cuff with asymmetric pressure Ischemic Trainer inflated to 150 mmHg SHAM BFR GROUP Double-chamber pneumatic cuff with asymmetric pressure Ischemic Trainer inflated to 20 mmHg	Control–6 20 mmHg BFR–6 150 mmHg BFR–6	Control–M3/F3 20 mmHg BFR–M3/F3 150 mmHg BFR–M3/F3	Control–36 (±9) 20 mmHg BFR–38 (±8) 150 mmHg BFR–38 (±6)	3 weeks (3 sessions a week)	3 weeks from the beginning of training
Ohta et al., 2003 [32]		Air torniquet at 180 mmHg Worn around proximal thigh	Control–22 BFR–22	Control–M12/F20 BFR–M13/F9	Control–30 (±9.7) BFR–28 (±9.7)	16 weeks (6 times weekly with different exercises throughout the program)	16 weeks from surgery
Jung et al., 2022 [33]	Korea	Smart Cuff at 40% of systolic blood pressure	Control–12 BFR–12	Control–M9/F3 BFR–M9/F3	Control–27.83 (±8.43) BFR–30.83 (±7.59)	12 weeks beginning 3 days post op (3 times a week with different exercises throughout the program)	12 weeks and 2 days from the beginning of training
Iversen et al., 2015 [27]	Norway	Delphi low pressure cuff with pressure starting at 130 mmHg and rising to 180 mmHg by 10th day of training	Control–12 BFR–12	Control–M7/F5 BFR–M7/F5	Control–29.8 (±9.3) BFR–24.9 (±7.4)	14 days with occlusion if in intervention group + 2 days without stimulus Beginning at 2 days post-surgery (2 times a day)	16 days post-surgery

**Table 2 jcm-13-06265-t002:** Results for primary outcome (difference in thigh/femur muscle).

Author	Control (N)	Intervention (R)	Way Outcome Was Measured	Control Results	Intervention Results	*p* Value
Jack et al., 2023 (6 weeks) [28]	17	15	DEXA scan of thigh, change in lean mass/kg	−0.27 (0.03)	−0.09 (0.03)	<0.01
(12 weeks)				−0.12 (0.05)	0.00 (0.03)	<0.01
Hughes et al., 2019 [9]	14	14	USS of vastus lateralis, change in muscle thickness/cm	1.94 (0.44)	1.91 (0.39)	0.85
De Melo et al., 2022 [29]	12	12	Not recorded	Not recorded	Not recorded	Not recorded
Li et al., 2023 [30]	6	40% BFR–9 80% BFR–8	Muscle thickness/cm	4.32 (0.7)	40% BFR–5.23 (0.84) 80% BFR–6.26 (0.64)	N vs. 40% BFR < 0.05 (0.017) N vs. 80% BFR < 0.05 (0.000) 40% vs. 80% <0.05 (0.004)
Kacin et al., 2021 [31]		20 mmHg BFR–6 150 mmHg BFR–6	Muscle size (cross sectional area of quadriceps)/mm^2^	Not reported	20 mmHg BFR: −1.1(2.1%) 150 mmHg BFR: 5.0 (3%)	20 mmHg vs. 150 mmHg < 0.01
Ohta et al., 2003 [32]		22	Not recorded	Not recorded	Not recorded	Not recorded
Jung et al., 2022 [33]		12	Not recorded	Not recorded	Not recorded	Not recorded
Iversen et al., 2015 [27]		12	Cross sectional area of femur 40% from knee joint/cm^2^ Cross sectional area of femur 50% from knee joint/cm^2^	40%: −9.2 (0.8) 50%: −11.5 (0.7) Mean Change: −13.1 (1.0)	40%: −9.7 (1.0) 50%: −13.7 (0.9) Mean Change: −13.8 (1.1)	0.6265

## Data Availability

All data generated as part of this study has been included in the manuscript.

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
