# Peer review of "Blood Flow Restriction Training and Its Use in Rehabilitation After Anterior Cruciate Ligament Reconstruction: A Systematic Review and Meta-Analysis"

_jcm, 2024, doi:10.3390/jcm13206265_

Round 1

Reviewer 1 Report

Comments and Suggestions for Authors

Please see the enclosed comments in the attachment.

Author Response

Thank you for taking the time to review our paper and offer some very valuable comments. We have revised as per your suggestions and all changes are tracked using the track changes facilities in Microsoft Word. Below is a point-by-point response:

Comment 1: The introduction could benefit from expanding a little more on the standard rehabilitation methods for ACLR and why they fall short. While muscle atrophy and deficits in quadriceps strength are mentioned, providing a brief overview of typical rehabilitation outcomes and challenges would be helpful. This would give a broader

audience a better context.

Response 1: The introduction highlights that persistent strength deficits are one of the main issues following ACLR and as such are the focus of this review.

Comment 2: While the justification for undertaking the review is generally sound, it could be made more explicit. The rationale for conducting an updated review is based on the growth of literature and the popularity of BFRT. It could benefit from emphasizing specific gaps in previous reviews, such as a lack of standardized protocols or inconsistent outcome measures. This would strengthen the case for why a new synthesis is needed. On the other hand, in this particular review, there is also a huge variety of protocols for applying BFR. Please somehow address this issue.

Response 2: Our study already mentions most of what is said here in Lines 70-86. However, we have also added sentences (Lines 91-99) to further emphasize the need for this review.

Comment 3: The choice of outcomes (Lysholm and IKDC knee scores) is mentioned, but more details could be provided on why these specific outcomes were chosen and how they add to the literature. A brief explanation of their relevance to functional recovery following ACLR would strengthen this section.

Response 3: A brief rationale has been provided in lines 118 – 122.

Comment 4: Regarding the aim of the review, the phrasing can be made more concise, and the objective could be explicitly tied to PICO in the structure of the research question.

Response 4: The changes have been made and an explicit PICO structure has been used to convey this (line 89-122).

Methods

Comment 1: The exact search string for each database needs to be included (it only partially describes the search terms). The authors should provide the whole search strategy as a supplemental file.

Response 1:  Clarification has been included on lines 133-134, the same search string was used across all three databases

Comment 2: Registration in a public repository (like PROSPERO) is not mentioned, and the text confirms the study was not registered. According to PRISMA, registration is recommended to avoid bias and transparency concerns. Why didn’t the authors register the review protocol? Please think about elaborating it in the Discussion and address the need for registering the protocols:

Prill R, Hakam H, Karlsson J, Ramadanov N, Alfuth M, Królikowska A (2024) Structured success: Study protocols and preregistration in orthopaedics, sports medicine and rehabilitation. Knee Surg Sports Traumatol Arthrosc 32:1065-1070.

Response 2: To explain this, we have added the following sentences in the limitations section: “A potential limitation of our systematic review is that we did not register the study into an appropriate public repository. This was an oversight on our part, and it is something that we also recommend to avoid potential bias. Having the review registered allows for any deviations from the original protocol to be documented.” Lines 579-582.

Comment 3: Documentation of date restrictions needs to be included. Also, be clear about languages—do you mean you included all languages? At which stage did you check that the text is available in English?

Response 3:  Clarification about date restrictions have been included in line 149-151 in section 2.3. Also language inclusions are mentioned in this section, Lines 139-153.  We write that following verification of selected articles; full text screening took place to ensure that all inclusion and exclusion criteria had been met. It is at this stage that texts that were not available in English would have been excluded.

Comment 4: More details on the data extraction forms or templates must be provided.

Response 4: Data extraction forms for all outcomes have been included in the Supplementary Materials.  

Comment 5: Information about whether the authors tried to contact study authors to resolve unclear data needs to be included.

Response 5: We have now included the statement: “Authors were contacted for missing data, but no responses were received during the time frame of this review”. Lines 186-187.

Comment 6: The description of the data extracted is quite thorough, including interventions, outcomes, and participant characteristics. However, as the protocol was not preregistered, we don’t know if all prespecified outcomes predefined in the protocol were collected.

Response 6: We understand that the protocol was not pre-registered, however, all pre-specified outcomes were collected and reported. In other words, we did not deviate from the original protocol.

Comment 7: The Risk of Bias 2 (RoB-2) tool is mentioned, which is in line with PRISMA recommendations; however, there is no mention of how the risk of bias results were used in interpreting the results or if it impacted the meta-analysis.

Response 7: We have added the following sentence in the limitations section to describe this: “Whilst the risk of bias analysis did not prevent our outcomes reporting or performing me-ta-analysis with pooled data from the outcome, the interpretation of the findings was severely affected. A high risk of bias means that any significant effects described in the me-ta-analysis must be interpreted with caution.” Lines 546-549.

Comment 8: A clear rationale for using a random effects model over a fixed effects model is not explicitly provided, though it’s commonly understood.

Response 8: We have provided the following rationale in Section 2.6 of the Methods: “A random effects model was chosen over fixed effects due to the heterogeneity in the studies where the random effects model can cope with both within-study and between-study variances.” Lines 207-210.

Comment 9: There is no mention of additional analyses like subgroup or sensitivity analysis, which are essential for exploring heterogeneity.

Response 9: We have explained in the Results in Section 3.4 that: “Subgroup or sensitivity analyses were not performed due to the high risk of bias which impacts on all data interpretation in this systematic review.” Lines 313-451.

Comment 10: When preparing the methods section, there are some articles that would be helpful:Prill R, Królikowska A, Becker R, Karlsson J (2023) Why there is a need to improve evaluation standards for clinical studies in orthopaedic and sports medicine. Knee Surgery, Sports Traumatology, Arthroscopy 31:4-5.

Prill R, Królikowska A, de Girolamo L, Becker R, Karlsson J (2023) Checklists, risk of bias tools, and reporting guidelines for research in orthopedics, sports medicine, and rehabilitation. Knee Surgery, Sports Traumatology, Arthroscopy 31:3029-3033.

Response 10: We believe that our systematic review closely follows the general guidelines in these studies.

Discussion

Comment 1: The discussion does not compare this review's findings with those of previous systematic reviews or relevant studies on BFRT and ACLR rehabilitation. A deeper comparison with existing research, especially the 2023 systematic review mentioned in the introduction, could strengthen the findings by situating them within the broader body of evidence.

Response 1: We have made reference to the previous systematic review and how there is still a need for more standardised protocols when assessing for BFRT and its efficacy following ACLR. Lines 506-530.

Comment 2: While the section hints at BFRT being useful in ACLR rehabilitation, there is no clear articulation of the clinical significance of the findings. For example, how would BFRT impact clinical practice or patient care compared to standard rehabilitation?

Response 2: Additions made to the introduction should help with the understanding about the use of BFRT within clinical practice.

Comment 3: The discussion acknowledges heterogeneity in protocols and outcome measurements but does not dive deeply into why this heterogeneity exists. It would be helpful to explore the possible sources of heterogeneity, such as differences in occlusion pressures, exercise intensity, patient populations, or the timing of BFRT post-surgery.

Response 3: Table 1 and section 3.2 of this paper allude to the heterogenous nature of the literature in this field, and it can be seen that whilst there are early signs of movement in the right direction with regards to cuff type and AOP, significant heterogeneity remains with regards to the reporting of primary outcomes such as muscle size and strength. Lines 515-530.

Comment 4: There is no mention of potential adverse effects or risks associated with BFRT, such as the risk of venous thrombosis, patient discomfort, or improper cuff use. These risks should be acknowledged and weighed against the potential benefits. Read also:

Królikowska A, Kusienicka K, Lazarek E, Oleksy Ł, Prill R, Kołcz A, et al. (2023) A Randomized, Double-Blind Placebo Control Study on the Effect of a Blood Flow Restriction by an Inflatable Cuff Worn around the Arm on the Wrist Joint Position Sense in Healthy Recreational Athletes. J Clin Med.

Response 4: We feel that this is out of the scope of this systematic review as we aimed to weigh up the evidence for BFRT and whether it leads to improved outcomes. We feel that the risks associated with BFRT are therefore out of scope but we do recognise that all clinical interventions are not without risk and within the wider context, BFRT does not appear to be any more risky than traditional strength training and has also been demonstrated to be more comfortable. In addition, the concern about DVT is very valid due to the potential consequences, however, these concerns are shown to be more of an issue in medically unoptimised cohorts who already suffer from comorbidities. The literature surrounding the risks faces similar issues that we have come across with this study in that it is heterogenous, thus making it difficult to draw conclusions. See example:

https://www.ncbi.nlm.nih.gov/pmc/articles/PMC8975582/

https://pubmed.ncbi.nlm.nih.gov/21410544/

https://www.frontiersin.org/journals/physiology/articles/10.3389/fphys.2022.808622/full

https://www.ncbi.nlm.nih.gov/pmc/articles/PMC6530612/

Therefore, we have not discussed this as it is a lengthy discussion and will detract from the main issues of BFRT research, where heterogeneity is a serious problem that limits the validity of this method.

Comment 5: While future research directions are touched upon, more specific suggestions could be made, such as investigating the optimal occlusion pressures, exercise intensities, or timing of BFRT in relation to surgery. Additionally, there is no mention of the need for multi-centre trials or studies on diverse patient populations (e.g., age groups and activity levels).

Response 5: We have expanded on the directions for future research in the conclusions section, highlighting the need for a broader range of patient demographics.

Reviewer 2 Report

Comments and Suggestions for Authors

An important topic of BFRT in ACL rehabilitation is reviewed here, but it has significant methodological limitations. A systematic review is performed by at least two authors; this is below the recommended minimum of three reviewers considered by the PRISMA standards. Concerns also arise on the rigor and possible biases inherent in the study selection processes, data extraction, and evaluation of bias. The involvement of at least three reviewers could make it more comprehensive and reliable. There are several useful insights arising from this review; it does need to be stated, however, that across these included studies, there is quite a difference in the BFRT methods and measures of outcome, and that is really a huge challenge to meta-analysis. It would be more desirable if an additional explanation of the data extraction techniques and a more extensive analysis of the biases could be included in the work. 

Comments on the Quality of English Language

The quality of the English language is generally clear and comprehensible, however, careful proofreading or review by a native English speaker would enhance the overall clarity and professionalism of the manuscript.

Author Response

Thank you for taking the time to review our paper and offer some very valuable comments. We have revised as per your suggestions and all changes are tracked using the track changes facilities in Microsoft Word. Below is a point-by-point response:

Comment 1: An important topic of BFRT in ACL rehabilitation is reviewed here, but it has significant methodological limitations. A systematic review is performed by at least two authors; this is below the recommended minimum of three reviewers considered by the PRISMA standards. Concerns also arise on the rigor and possible biases inherent in the study selection processes, data extraction, and evaluation of bias. The involvement of at least three reviewers could make it more comprehensive and reliable. Lines 579-585.

Response 1: Thank you for recognising the importance of the topic. Form what we can find, the PRISMA guidelines do not specify a minimum number of reviewers for a systematic review. However, most authors recommend at least 2 and some authors recommend a third reviewer to refer to if there are discrepancies between the first two. What is the norm in the literature is that the majority of systematic reviews published involved 2 reviewers. Hence, our systematic review does not pose any more bias than the majority of the currently published systematic reviews. For our systematic review, both authors reviewed studies independently of each other to avoid risk of bias.

Comment 2: There are several useful insights arising from this review; it does need to be stated, however, that across these included studies, there is quite a difference in the BFRT methods and measures of outcome, and that is really a huge challenge to meta-analysis. It would be more desirable if an additional explanation of the data extraction techniques and a more extensive analysis of the biases could be included in the work.

Response 2: The differences in BFRT methods and outcomes from the included studies is not due to our data extraction techniques because we simply relied on the published data in these studies but they are due to the variety of methods used and the wide differences in data reported. This is a significant issue we wanted to highlight that very few studies used the same intervention techniques and the same outcome measures and hence comparisons are difficult to make. This means that future high quality studies need to be performed with agreed outcome measures so that interventions can be analysed constructively. 

Comment 3: The quality of the English language is generally clear and comprehensible, however, careful proofreading or review by a native English speaker would enhance the overall clarity and professionalism of the manuscript.

Response 3: Both authors are “native English speakers” and have completed all of their formal education in world leading Institutions in English. Nonetheless, we have carefully completed proofreading to ensure clarity and professionalism of the manuscript. 

Reviewer 3 Report

Comments and Suggestions for Authors

Dear Authors,

I have reviewed your manuscript, and it demonstrates the potential of blood flow restriction training in ACL rehabilitation. To further improve the quality of your work, I have the following suggestions for enhancement:

The use of the RoB-2 tool to assess the risk of bias is appropriate for RCTs. However, the reporting of these biases, particularly unclear randomization protocols and inconsistent intervention adherence, could be expanded to better explain their impact on the study’s overall conclusions. A clearer discussion of how these biases influence the interpretation of the findings and how they affect the narrative synthesis would enhance the clarity and robustness of the paper.

The meta-analyses for the secondary outcomes (e.g., IKDC scores, I² = 49%) indicate moderate heterogeneity, which requires further discussion. I recommend exploring potential sources of this heterogeneity, such as differences in protocols or patient characteristics. Additionally, conducting a sensitivity analysis could help assess the robustness of the findings and strengthen the interpretation.

The discussion section would benefit from a deeper analysis of why some studies reported significant benefits of BFRT while others did not. Specifically, potential reasons such as variations in BFRT protocols, patient populations, and outcome measures should be explored to provide clearer insights into the observed differences in findings.

Improve the clarity of Figures 3 and 4 to enhance their visual representation.

While the exclusion of athletic populations is justified to focus on the general population, it may be helpful to briefly mention in the limitations section how this exclusion affects the generalizability of the findings to athletes, given they represent a common ACLR demographic.

Author Response

Thank you for taking the time to review our paper and offer some very valuable comments. We have revised as per your suggestions and all changes are tracked using the track changes facilities in Microsoft Word. Below is a point-by-point response:

Comment 1: I have reviewed your manuscript, and it demonstrates the potential of blood flow restriction training in ACL rehabilitation.

Response 1: Thank you for taking the time to review our work and offer these recommendations to improve the manuscript.

Comment 2: The use of the RoB-2 tool to assess the risk of bias is appropriate for RCTs. However, the reporting of these biases, particularly unclear randomization protocols and inconsistent intervention adherence, could be expanded to better explain their impact on the study’s overall conclusions. A clearer discussion of how these biases influence the interpretation of the findings and how they affect the narrative synthesis would enhance the clarity and robustness of the paper.

Response 2: We have added the following sentences in the limitations section: “Whilst the risk of bias analysis did not prevent our outcomes reporting or performing meta-analysis with pooled data from the outcome, the interpretation of the findings was severely affected. A high risk of bias means that any significant effects described in the me-ta-analysis must be interpreted with caution.” Lines 544-547.

Comment 3: The meta-analyses for the secondary outcomes (e.g., IKDC scores, I² = 49%) indicate moderate heterogeneity, which requires further discussion. I recommend exploring potential sources of this heterogeneity, such as differences in protocols or patient characteristics. Additionally, conducting a sensitivity analysis could help assess the robustness of the findings and strengthen the interpretation.

Response 3: Please refer to response 3 at the beginning of the methods section.

Comment 4: The discussion section would benefit from a deeper analysis of why some studies reported significant benefits of BFRT while others did not. Specifically, potential reasons such as variations in BFRT protocols, patient populations, and outcome measures should be explored to provide clearer insights into the observed differences in findings.

Response 4: Please refer to response 3 at the beginning of the methods section.

Comment 5: Improve the clarity of Figures 3 and 4 to enhance their visual representation.

Response 5: Figures 3 and 4 are standard output files from Revman but we have made them larger so that they are more easily readable.

Comment 6: While the exclusion of athletic populations is justified to focus on the general population, it may be helpful to briefly mention in the limitations section how this exclusion affects the generalizability of the findings to athletes, given they represent a common ACLR demographic.

Response 6: Thank you, we have added the following sentence in the limitations section: “In addition, the exclusion of athletes to focus on recovery in the general population may mean that our results are not generalizable to this group of people who are at most risk of ACLR.” Lines 582-585.

Round 2

Reviewer 1 Report

Comments and Suggestions for Authors

The authors addressed all the comments.

Reviewer 2 Report

Comments and Suggestions for Authors

This systematic review yields informative findings related to the application of blood flow restriction training during ACL rehabilitation and henceforth addresses an important clinical issue. Study design is appropriate, and the methods are well-identified in a PRISMA-guided way. The results are well-presented; discussion might be further developed with a greater focus on the clinical implications of the statistical findings. Overall, this is a well-conceptualized study. It contributes significantly to the growing literature about BFRT and points toward a clear need for future research with respect to standardization regarding both protocols and outcome measures.